# Evaluating the Impact of Different Hypercaloric Diets on Weight Gain, Insulin Resistance, Glucose Intolerance, and its Comorbidities in Rats

**DOI:** 10.3390/nu11061197

**Published:** 2019-05-28

**Authors:** Bernardete F. Melo, Joana F. Sacramento, Maria J. Ribeiro, Claudia S. Prego, Miguel C. Correia, Joana C. Coelho, Joao P. Cunha-Guimaraes, Tiago Rodrigues, Ines B. Martins, Maria P. Guarino, Raquel M. Seiça, Paulo Matafome, Silvia V. Conde

**Affiliations:** 1CEDOC, NOVA Medical School, Faculdade de Ciências Médicas, 1150-082 Lisboa, Portugal; Bernardete.melo@nms.unl.pt (B.F.M.); joana.sacramento@nms.unl.pt (J.F.S.); mj.rfribeiro@gmail.com (M.J.R.); claudia.prego@nms.unl.pt (C.S.P.); mc.correia@campus.fct.unl.pt (M.C.C.); joana.indias@gmail.com (J.C.C.); jp.guimaraes@campus.fct.unl.pt (J.P.C.-G.); iib.martins@campus.fct.unl.pt (I.B.M.); maria.guarino@ioleiria.pt (M.P.G.); 2Instituto de Investigação Clínica e Biomédica de Coimbra (iCBR), Faculdade de Medicina de Coimbra, 3000-548 Coimbra, Portugal; tiagodarodrigues@gmail.com (T.R.); rmfseica@gmail.com (R.M.S.); paulomatafome@gmail.com (P.M.); 3Escola Superior de Saúde de Leiria- Instituto Politécnico de Leiria, 2410-197 Leiria, Portugal; 4Instituto Politécnico de Coimbra, Coimbra Health School (ESTeSC), Department of Complementary Sciences, 3046-854 Coimbra, Portugal

**Keywords:** type 2 diabetes, obesity, metabolic syndrome, insulin sensitivity, glucose tolerance, weight gain, hypertension, adipose tissue, animal models

## Abstract

Animal experimentation has a long history in the study of metabolic syndrome-related disorders. However, no consensus exists on the best models to study these syndromes. Knowing that different diets can precipitate different metabolic disease phenotypes, herein we characterized several hypercaloric rat models of obesity and type 2 diabetes, comparing each with a genetic model, with the aim of identifying the most appropriate model of metabolic disease. The effect of hypercaloric diets (high fat (HF), high sucrose (HSu), high fat plus high sucrose (HFHSu) and high fat plus streptozotocin (HF+STZ) during different exposure times (HF 3 weeks, HF 19 weeks, HSu 4 weeks, HSu 16 weeks, HFHSu 25 weeks, HF3 weeks + STZ) were compared with the Zucker fatty rat. Each model was evaluated for weight gain, fat mass, fasting plasma glucose, insulin and C-peptide, insulin sensitivity, glucose tolerance, lipid profile and liver lipid deposition, blood pressure, and autonomic nervous system function. All animal models presented with insulin resistance and dyslipidemia except the HF+STZ and HSu 4 weeks, which argues against the use of these models as metabolic syndrome models. Of the remaining animal models, a higher weight gain was exhibited by the Zucker fatty rat and wild type rats submitted to a HF diet for 19 weeks. We conclude that the latter model presents a phenotype most consistent with that observed in humans with metabolic disease, exhibiting the majority of the phenotypic features and comorbidities associated with type 2 diabetes in humans.

## 1. Introduction

Over the last few years, the incidence and prevalence of metabolic diseases such as obesity, type 2 diabetes (T2D) and metabolic syndrome has increased dramatically, recently being highlighted as a worldwide epidemic, with a high socioeconomic impact [1]. This increase emphasizes the urgent need to understand the causes and mechanisms underlying the onset and progression of these metabolic disorders, aiming to develop new therapeutic strategies to prevent or halt the progressive rise in incidence of these disorders. This growing epidemic highlights the need for further experimental research, enabled by the availability of animal models that closely mimic the metabolic and cardiovascular phenotype exhibited by humans.

Pathological weight gain is the major cause of both metabolic and cardiovascular diseases [1,2,3], with 1 in 5 deaths in the world today associated with obesity [3]. In fact, in 2016, more than 1.9 billion adults were overweight and from these, more than 650 million were obese [3]. Obesity is defined by the World Health Organization (WHO) as an excessive adipose tissue accumulation sufficient to impair health [3,4]. It is characterized by excessive central visceral adiposity that contributes to a chronic increase in circulating free fatty acids and the resulting metabolites that will activate various signaling cascades, interfering with insulin signaling and β-cell function and increasing circulatory disease and cancer risk [4,5].

Type 2 diabetes (T2D) accounts for over 90–95% of all diabetes, being a complex multifactorial metabolic disorder influenced by lifestyle, environmental, and genetic risk factors. In 2017, 425 million people worldwide had diabetes, a number that continues to increase and which, by 2045, is anticipated to affect 629 million people [6,7]. T2D is characterized by impaired glucose homeostasis with insulin resistance and β-cell dysfunction, the primary trait induced by obesity being insulin resistance in metabolic tissues (adipose, hepatic, and muscular tissues). This peripheral insulin resistance induces pancreatic β cells to secrete more insulin, leading to hyperinsulinemia, which often leads to β cell depletion and sustained hyperglycemia in T2D [7].

Metabolic syndrome is defined by a cluster of metabolic abnormalities that increase the risk of coronary heart disease, other forms of cardiovascular diseases, and T2D. Its main pathological characteristics are obesity, insulin resistance, raised fasting plasma glucose, hypertension, and atherogenic dyslipidemia (elevated serum triacylglycerols and apolipoprotein B, increased small low-density lipoprotein (LDL) particles, and a reduced level of high-density lipoprotein (HDL)) [8,9,10].

One of the mechanisms involved in the pathogenesis of metabolic diseases and its comorbidities is autonomic dysfunction. Obesity, T2D, and metabolic syndrome are all associated with an increased sympathetic and decreased parasympathetic drive [11,12], with previous studies showing an increase of serum noradrenaline, noradrenaline excretion, renal and heart and muscle sympathetic nervous system activity, and heart rate variability indexes [13,14,15,16]. Autonomic dysfunction has been linked to target organ damage, such as diastolic dysfunction, ventricular hypertrophy or cardiac remodeling, as well as with insulin resistance, hyperinsulinemia, and dyslipidemia [13,15]. It has been postulated that prevention of autonomic dysfunction could be a strategy for reducing these comorbidities and that the assessment of autonomic function could be used as an early biomarker for metabolic diseases [17,18].

Several rodent models have been developed to study metabolic disturbances, particularly obesity and T2D. In general, T2D is induced by surgical, chemical, dietary or genetic manipulations, or by a combination of these and other techniques. However, the pathophysiological conditions of these types of models are far from the human characteristics of disease. Thus, hypercaloric, hyperlipidemic diets or the combination of both have been used to induce obesity and T2D as well as metabolic syndrome in animals [19,20,21,22]. However, a lack of consensus exists on the best diets and time of exposure to these diets used to promote alterations in metabolic parameters, such as insulin resistance, hyperinsulinemia, and glucose intolerance, among others, as well as associated metabolic comorbidities, such as hypertension and non-alcoholic fatty liver disease.

The present investigation evaluates the impact of different hypercaloric diets and different times of exposure to diets on weight gain, insulin resistance, glucose intolerance, as well as autonomic dysfunction and consequences hypertension, and lipid deposition in the liver as an indication of non-alcoholic fatty liver disease. Additionally, we compared these models with a genetic model, the Zucker fatty diabetic rat, which is a reference model for obesity and T2D. Moreover, we discuss the outcomes for these different animal models, with the aim of identifying the most appropriate model(s) to investigate physiopathological mechanisms and therapeutic strategies for metabolic diseases.

## 2. Methods

### 2.1. Diets and Animal Care

Experiments were performed in 8–9-weeks-old male Wistar rats (200–300 g) obtained from the animal house of NOVA Medical School, Faculty of Medical Sciences, New University of LisbonUniversidade Nova de Lisboa, Lisbon, Portugal, except for the male Zucker diabetic fatty (ZDF) animals and their respective controls Zucker lean that were acquired with 6-weeks-old from Charles River (Paris, France) and maintained for 2 weeks in quarantine. After randomization, the hypercaloric diet animals were assigned to one of six groups: (1) The 3 weeks high-fat diet-fed (HF) group, fed with a 45% fat diet (45% fat + 35% carbohydrate + 20% protein; for detailed composition, see Appendix A; Mucedola, Milan, Italy); (2) the 19 weeks HF group, fed with 60% fat diet (61.6% fat + 20.3% carbohydrate + 19.1% protein; for detailed composition see Appendix A; Test Diets, Missouri, USA); (3) the 4 weeks HF group, fed with a 45% fat diet (45% fat + 35% carbohydrate + 20% protein; Mucedola, Milan, Italy) and submitted to an injection of streptozotocin (25 mg/kg, i.p.), an antibiotic derived from Streptomyces achromogenes that causes β-cell damage [23], in the beginning of the 4th week of diet (HF+STZ group); (4) the high-sucrose diet-fed (HSu) group, fed with 35% (wt/vol.) sucrose (PanReac, Madrid, Spain) in drinking water for 4 weeks; (5) the HSu group fed with 35% (wt/vol.) sucrose (PanReac, Madrid, Spain) in drinking water for 16 weeks; and (6) a combined model of HF (61.6% fat + 20.3% carbohydrate + 19.1% protein; Test Diets, Missouri, USA) and HSu (35% (wt/vol.)) diet for 25 weeks. ZDF animals were fed with Purina 5008 (Formulab Diet 5008 and Formulab Diet 5008C33, for detailed composition, see Appendix A; Purina) that consisted of a mix of 23.6% protein, 14.8% lipid, 50.3% carbohydrates, 3.3% fiber, and the remaining minerals and vitamins. All control animals were fed a control diet (7.4% lipid and 75% carbohydrates, of which 4% were sugars and 17% protein; for detailed composition, see Appendix A; SDS RM3). All groups of hypercaloric diet animals and its aged-matched controls as well as Zucker diabetic rat and its lean controls of 17 weeks included 6–8 animals per group; Zucker diabetic rat of 23 weeks included 3 animals per group, and its lean control only one animal.

Animals were kept under temperature and humidity control (21 ± 1 °C; 55 ± 10% humidity) with a 12 h light/12 h dark cycle and were given ad libitum access to food and water. Body weight was monitored weekly, and energy and liquid intake were monitored daily. Insulin sensitivity and glucose tolerance were also evaluated over the experimental period.

At a terminal experiment, animals were anesthetized with sodium pentobarbitone (60 mg/kg, i.p), and catheters were placed in the femoral artery for arterial blood pressure measurement. Afterwards, blood was collected by heart puncture for serum and plasma quantification of mediators and the liver, soleus, and gastrocnemius and the adipose tissue pads as the visceral, perinephric, epidydimal, and subcutaneous fat were then rapidly collected, weighted, and stored at −80 °C for further analysis. Laboratory care was in accordance with the European Union Directive for Protection of Vertebrates Used for Experimental and Other Scientific Ends (2010/63/EU). Experimental protocols were approved by the NOVA Medical School/Faculdade de Ciências Médicas Ethics Committee and by Portuguese DGAV.

### 2.2. Insulin Tolerance Test

Insulin sensitivity was evaluated using insulin tolerance test (ITT) after overnight fasting as previously described [24]. Briefly, fasting blood glucose was measured and immediately followed by an insulin bolus (100 mU/kg), administered via the tail vein. Subsequently, the decline in plasma glucose concentration was measured over a 15 min period.

### 2.3. Oral Glucose Tolerance Test

Glucose tolerance was evaluated using oral glucose tolerance test (OGTT) after overnight fasting as described by Sacramento et al. [25]. Briefly, fasting blood glucose was measured and immediately followed by administration of a saline solution of glucose (2 g/kg, VWR Chemicals, Leuven, Belgium), by gavage. Blood glucose levels were measured at 15, 30, 60, 120, and 180 min.

### 2.4. Quantification of Biomarkers: Plasma Insulin, C-Peptide, Lipid Profile, and Catecholamines

Insulin and C-peptide concentrations were determined with an enzyme-linked immunosorbent assay kit (Mercodia Ultrasensitive Rat Insulin ELISA Kit and Mercodia Rat C-peptide ELISA Kit, respectively, Mercodia AB, Uppsala, Sweden). Catecholamines were measured in plasma and in homogenized adrenal medulla samples as previously described [25].

The lipid profile was assessed using a RANDOX kit (RANDOX, Irlandox, Porto, Portugal) as described by Sacramento et al. [23].

### 2.5. Blood Pressure Evaluation

Mean arterial pressure (MAP) was determined by catheterization of the femoral artery. The catheter was connected to a pressure transducer (−50, +300 mmHg) and amplifier (Emka Technologies, Paris, France). MAP was calculated using the values of systolic blood pressure (SBP) and diastolic blood pressure (DBP) by the Iox 2.9.5.73 software (Emka Technologies, Paris, France). To calculate a mean arterial pressure, double the diastolic blood pressure and add the sum to the systolic blood pressure. Then divide by 3. MAP = (SBP + 2 (DBP))/3.

### 2.6. Lipid Deposition in the Liver

Lipid deposition in the liver was quantified by an extraction method described by Folch [26]. Briefly, 0.5 g of liver samples were homogenized in 3 ml of Folch solution. The samples were shaken and filtrated to a test tube and the process repeated twice. After, 2 ml of NaCl 0.73% were added to the filtrated samples and samples were left in rest overnight. Afterwards, the organic phase was collected and transferred to a petri dish. To the other phase was added 2.5 mL of Folch solution: NaCl 0.58% (80:20). The process was repeated, and all the organic phase collected was left to dry for 24 h before weighing the petri dish, for determination of total lipids in the sample. The results were expressed as a percentage of lipids per total weight of homogenized liver.

### 2.7. Western Blot Analysis

Visceral adipose tissue (100 mg) was homogenized in Zurich (10 mM Tris-HCl, 1 mM EDTA, 150 mM NaCl, 1% Triton X-100, 1% sodium cholate, 1% SDS) with a cocktail of protease inhibitors (trypsin, pepstatin, leupeptin, aprotinin, sodium orthovanadate, phenylmethylsulfonyl fluoride (PMSF)), and samples were centrifuged (Eppendorf, Madrid, Spain) and supernatant was collected and frozen at −80 °C until further use.

Samples of the homogenates (50 μg) and the prestained molecular weight markers (Precision, BioRad, Madrid, Spain) were separated by sodium dodecyl sulfate polyacrylamide gel electrophoresis (10% with a 5% concentrating gel) under reducing conditions and electrotransferred to polyvinylidene difluoride membranes (0.45 μM, Millipore, Spain). After blocking for 1 h at room temperature with 5% nonfat milk in Tris-buffered saline, pH 7.4 containing 0.1% Tween 20 (TTBS) (BioRad, Spain), the membranes were incubated overnight at 4 °C with the primary antibodies against glucose transporter type 4 (GLUT4; 1:200, Abcam, Cambridge, UK), insulin receptor (IR; 1:200, Santa Cruz Biotechnology, Madrid, Spain), phosphorylated insulin receptor (p-Tyr1361, 1:500, Abcam, Cambridge, UK), Protein Kinase B (Akt; 1:1000, Cell Signaling, Leiden, The Netherlands) and phophorylated Akt (p-Ser473, 1:1000, Cell Signaling, Leiden, The Netherlands). The membranes were washed with Tris-buffered saline with Tween (TBST) (0.1%) and incubated with mouse anti-goat (1:2000) or goat anti-mouse (1:2000) in TBS and developed with enhanced chemiluminescence reagents in accordance with the manufacturer’s instructions (ClarityTM Western ECL substrate, Hercules, CA, USA). Intensity of the signals was detected in a Chemidoc Molecular Imager (Chemidoc; BioRad, Madrid, Spain) and quantified using Image Lab software (BioRad). The membranes were re-probed and tested for Calnexin (1:1000, SICGEN, Cantanhede, Portugal) immunoreactivity (bands in the 85 kDa region) to compare and normalize the expression of proteins with the amount of protein loaded.

### 2.8. Evaluation of Autonomous Nervous System

The balance between the sympathetic and parasympathetic components of the autonomic nervous system was made by calculating the sympathetic nervous system (SNS) and parasympathetic nervous system (PNS) indexes computed in Kubios HRV software (www.kubios.com). The SNS index in Kubios is based on Mean heart rate, Baevsky’s stress index, and low frequency power expressed in normalized units and the PNS index which is based on the mean intervals between successive heartbeats (RR intervals), the root mean square of successive RR interval differences (RMSSD) and high frequency power expressed in normalized units. Heart rate and RR intervals were obtained using Iox 2.9.5.73 software (Emka Technologies, Paris, France), with an acquisition frequency of 500 Hz.

### 2.9. Statistical Analysis

Data were evaluated using GraphPad Prism Software, version 6 (GraphPad Software Inc., San Diego, CA, USA) and presented as mean values with SEM. The significance of the differences between the mean values was calculated by one- and two-way ANOVA with Bonferroni multiple comparison test. Differences were considered significant at *p* < 0.05.

## 3. Results

### 3.1. Weight Gain, Fat Mass Depots, and Lipid Profile in Animal Models of Type 2 Diabetes and Obesity

In Figure 1A, we depict the growth curves of the animal models of obesity and T2D studied and their correspondent age-matched controls and in where a clear increase can be noted in weight gain in almost all disease models. When the data are plotted as the increase in grams per day, the disease models presented an increased weight gain when compared to their correspondent age-matched controls (Figure 1B, HF 3 weeks = 1.43 ± 0.28; HF 19 weeks = 4.10 ± 0.30; HF+STZ = 3.45 ± 0.17; HSu 4 weeks = 1.31 ± 0.10; HSu 16 weeks = 0.68 ± 0.13; HFHSu 25 weeks = 2.60 ± 0.19; Zucker 17 weeks = 26.23 ± 1.27 Zucker 23 weeks = 19.12 ± 3.83; CTL 3 weeks = 0.82 ± 0.17; CTL 19 weeks = 2.66 ± 0.16; CTL for HF+STZ group = 1.12 ± 0.33; CTL 4 weeks = 0.82 ± 0.17; CTL 16 weeks = 0.54 ± 0.10; CTL 25 weeks = 1.43 ± 0.07; lean 17 weeks = 16.21 ± 0.81; lean 23 weeks = 13.18 ± 0.00 g/day). Only the sucrose-rich diet for 16 weeks did not produce a significant increase in weight gain in comparison to the controls. When comparing HF 19 weeks with all the disease models, induced by hypercaloric diets, it was observed that this group presented a more pronounced weight gain that was only exceeded by the genetic model, the Zucker rat, with 17 and 23 weeks (Figure 1B).

Table 1 depicts the amount of fat mass of the animal models of obesity and T2D studied and their correspondent age-matched controls. All disease models, except the HF+STZ and HSu 4 weeks, exhibited a significant increase in the total amount of fat when compared to age-matched controls. Furthermore, an increase was also observed in the amount of perienteric, epidydimal, and perinephric depots, except for the models described above, HF+STZ and HSu 4 weeks (Table 1). As can be seen in Table 1, when comparing all disease models, the HF 19 weeks group was the animal model that exhibited a higher amount of total fat as well as higher amount of all adipose tissue depots (Table 1).

When comparing the levels of cholesterol, HDL, LDL, and triglycerides, it was possible to identify the disease models that presented alterations in lipid profile, in comparison to their correspondent age-matched controls (Table 1). HF 19 weeks, HSu 4 weeks, and Zucker 17 weeks groups showed increased levels of total cholesterol and HF 3 weeks, HF 19 weeks, Hsu 4 weeks, HSu 16 weeks, HFHSu 25 weeks, and Zucker 17 weeks showed increased levels of triglycerides. Further, HDL levels were decreased in HF 3 weeks, HF 19 weeks, HSu 16 weeks, and Zucker 17 weeks, and LDL levels were increased in the HF 19 weeks, HSu 16 weeks, HFHsu 25 weeks, and Zucker 17 weeks groups. The HF 19 weeks and Zucker 17 weeks groups exhibited dysfunctional levels in all the parameters of the lipid profile evaluated.

### 3.2. Effect of Hypercaloric Diets and Genetic Deletion of Leptin Receptors on Glucose Metabolism in Animal Models of Obesity and Type 2 Diabetes

Figure 2A represents the effect of hypercaloric diets and genetic deletion of leptin receptors on fasting glycemia in animal models of obesity and T2D and their correspondent age-matched controls. All the disease groups, except the HSu 16 weeks, presented a significant increase in fasting glycemia (% increase glycemia: 15% HF 3 weeks, 13% HF 19 weeks, 33% HF+STZ, 25% HSu 4 weeks, 17% HF+HSu 25 weeks, 154% Zucker 17 weeks, 201% Zucker 23 weeks). Note, however, that even though fasting glycemia levels increased in all models, only the genetic model presented values above 126 mg/dl, the reference value for diabetes diagnosis.

As depicted in Figure 2B, insulin resistance was observed in all disease models, as the constant rate for glucose disappearance (K_ITT_) significantly decreased when compared to their correspondent age-matched controls, except for the HF+STZ group, (HF 3 weeks = 2.07 ± 0.19; HF 19 weeks = 1.97 ± 0.29; HF+STZ = 5.01 ± 0.18; HSu 4 weeks = 2.45 ± 0.25; HSu 16 weeks = 1.79 ± 0.19; HFHSu 25 weeks = 1.82 ± 0.13; Zucker 17 weeks = 1.55 ± 0.47 Zucker 23 weeks = 1.59 ± 0.33; CTL 3 weeks = 4.56 ± 0.41; CTL 19 weeks = 4.32 ± 0.26; CTL for HF+STZ group = 4.17 ± 0.31; CTL 4 weeks = 4.56 ± 0.41; CTL 16 weeks = 5.29 ± 0.51; CTL 25 weeks = 4.73 ± 0.30; lean 17 weeks = 4.46 ± 0.49; lean 23 weeks = 4.22 ± 0.00% glucose/min). It can be clearly seen that a more pronounced insulin resistance was obtained with longer exposures to hypercaloric diets as well as with the genetic deletion of leptin receptors.

In agreement with these characteristics, Figure 2C represents the glucose tolerance depicted as glucose excursion curves (left panel) and as the area under the curve (AUC) obtained from the glucose excursion curves (right panel). Comparing the AUC of glucose excursion curves in the disease animal models with the AUC of their correspondent age-matched controls, it was observed that the groups HF 19 weeks, HF+STZ, HSu 16 weeks, HF+HSu 25 weeks, and Zucker 17 and 23 weeks presented glucose intolerance (HF 19 weeks = 24,406.29 ± 393.89; HF+STZ = 31,330.83 ± 1625.33; HSu 16 weeks = 21,502.75 ± 720.46; HF+HSu 25 weeks = 24,969.63 ± 635.89; Zucker 17 weeks = 56,526.00 ± 5115.70; Zucker 23 weeks = 69,514 ± 1414; CTL 19 weeks = 19,668 ± 380.34; CTL for HF+STZ = 23,427.83 ± 567.41; CTL 16 weeks = 20,689.80 ± 836.43; CTL 25 weeks = 21,986 ± 368.85; lean 17 weeks = 16,849 ± 616.37; lean 23 weeks = 16,208 ± 0.00 mg/dl*min). As expected, Zucker animals were the animal model that showed a more pronounced increase in fasting glycemia, higher insulin resistance, and higher glucose intolerance, followed by the diet-induced glucose dysmetabolism models, HF19 weeks and HFHSu 25 weeks groups, which also showed increases in all these characteristics, although with a lower magnitude.

In accordance with the whole-body insulin resistance, it was observed that all the disease animal models, except the HF+STZ model, exhibited increased fasting insulin values (Figure 2D). Insulin values increased significantly by 82%, 142%, 116%, 38%, 128%, 946%, and 441% in HF 3 weeks, HF 19 weeks, Hsu 4 weeks, Hsu 16 weeks, HFHSu 25 weeks, Zucker 17 weeks, and Zucker 23 weeks, respectively. The values of C-peptide, which measures endogenous insulin secretion, also significantly increased by 59%, 134%, 69%, 157%, 321%, and 102% in HF 3 weeks, HF 19 weeks, Hsu 4 weeks, HFHSu 25 weeks, Zucker 17 weeks, and Zucker 23 weeks, respectively. The absence of increase in insulin levels in the HF+STZ model is agreement with the lack of insulin resistance in this group. Note also that the Zucker animals were the groups that showed higher insulin and C-peptide levels, followed by HF 19 weeks and HFHSu 25 weeks, which agrees with the more profound alterations in glucose tolerance and whole-body insulin sensitivity.

In agreement with the alterations observed in glucose tolerance and whole-body insulin sensitivity, we observed profound alterations in the expression of proteins involved in insulin signaling on white adipose tissue from the HF 3, HSu 4, and HSu 16 weeks groups (Figure 3, please see Appendix A for complete Western blot gels (Appendix A)). The disease models tested showed decreased expression of the insulin receptor and its phosphorylated form by 26%, 18%, and 23% (for IR) and 34%, 33%, and 8% (for IR phosphorylated form) for HF 3, HSu 4 and HSu 16 weeks respectively. Further, the expression of Akt and its phosphorylated form decreased by 35% and 42% (for Akt) in HF 3 weeks and HSu 4 weeks, respectively, and by 26%, 43%, and 22% (for Akt phosphorylated form) in HF 3, HSu 4, and HSu 16 weeks respectively. As a glucose transporter, GLUT4 expression also decreased by 17%, 20%, and 22% in HF 3, HSu 4, and HSu 16 weeks, respectively. Note that the effects of diet on the expression of these proteins were higher for 4 weeks of HSu than for the 16 weeks, suggesting compensatory mechanisms to maintain cellular insulin sensitivity.

### 3.3. Comorbidities in the Distinct Models of Obesity and Type 2 Diabetes

Hypertension is one of the main illnesses associated with obesity and T2D (1-3). Figure 4 depicts mean arterial pressure levels for the animal models of obesity and T2D studied and their correspondent age-matched controls. All disease models exhibit increased levels of blood pressure when compared to their age-matched controls (HF 3 weeks = 127.30 ± 1.60; HF 19 weeks = 127.84 ± 6.81; HF+STZ = 119.08 ± 6.08; HSu 4 weeks = 127.04 ± 3.98; HFHSu 25 weeks = 121.74 ± 5.37; Zucker 17 weeks = 124.07 ± 6.34; Zucker 23 weeks = 129.80 ± 0.00; CTL 3 weeks = 98.36 ± 4.29; CTL 19 weeks = 95.38 ± 6.80; CTL for HF+STZ group = 89.62 ± 3.94; CTL 4 weeks = 98.36 ± 4.29; CTL 25 weeks = 82.89 ± 5.62; lean 17 weeks = 104.58 ± 3.92; lean 23 weeks = 115.10 ± 0.00 mmHg), HF 19 weeks (34%) and HFHSu 25 weeks (47%) being the animal models that exhibit higher increases.

Another common comorbidity of obesity and T2D is non-alcoholic fatty liver disease, which is characterized by increased lipid deposition in the liver. Figure 5A shows hematoxicilin and eosin images obtained from HF 19 weeks, HFHSu 25 weeks, and Zucker 23 weeks animals and their correspondent controls. All animals present an increase in lipid deposition with a marked hepatocellular micro and macrovesicular (black arrowhead) vacuolization, and presence of fibrosis (white arrowhead) However, it can be noted that lipid deposition in HF19 weeks and Zucker diabetic it is more representative at a microvesicular level, while HFHSu animals 25 weeks animals exhibit a higher percentage of macrovesicular vacuolization. Note also some degree of fibrosis in HF 19 weeks animals as well as in Zucker diabetic fatty 23 weeks. Figure 5B represents the percentage of lipids in the liver quantified by the Folch method, and it can be observed that all disease models showed higher lipid deposition when compared to their correspondent age-matched controls (% increase in lipid deposition HF 3 weeks = 88%; HF 19 weeks = 136%; HF+STZ = 86%; HSu 4 weeks = 33%; HSu 16 weeks = 31%; HFHSu 25 weeks = 48%; Zucker 17 weeks = 147%; Zucker 23 weeks = 123%)). All animals submitted to high fat diet (HF 3 weeks, HF 19 weeks, HF+STZ) and the Zucker animals were the groups that exhibited higher lipid deposition in the liver, with HSu animals showing a lower liver lipid deposition.

### 3.4. Autonomic Function in Obesity and Type 2 Diabetes

Sympathetic nervous system activation has been pointed out as one of the mechanisms contributing to cardiometabolic dysfunction in obesity and T2D. Herein, we have evaluated sympathetic and parasympathetic nervous activity using different methodologies: Through the evaluation of the SNS and PNS indexes (see Methods section) (Figure 6) and through the measurement of both circulating and adrenal medulla catecholamines (Table 2) in animal models of obesity and T2D studied. We have previously described that 3 weeks of HF diet and 4 weeks of HSu induced an overactivation of the sympathetic nervous system assessed by heart rate variability [25]. Herein we showed that hypercaloric diet disease models tested (HF 19 weeks and HFHSu 25 weeks), but not the genetic model, showed an increased SNS index in comparison to age-matched controls, suggesting an increased sympathetic activation (Figure 6B), without any alteration of the PNS. The absence of SNS activation in Zucker 17 weeks might be explained by the absence of the effect of leptin, a powerful sympathetic activator, in this animal model. In agreement with this increased sympathetic activation, hypercaloric disease models exhibited increased plasma catecholamine levels as well as increased catecholamines in adrenal medulla content when compared to their age-matched controls [Table 2 and Sacramento et al. [25]].

## 4. Discussion

In this study, we evaluated the effect of different hypercaloric diets and different exposure times to obesogenic diets as well as the genetic deletion of leptin receptors in terms of the key pathological characteristics of obesity and T2D, with the aim of identifying the most appropriate model(s) for the study dysmetabolism. We observed that all the disease models exhibited: (1) Increased weight gain; (2) increased amount of white adipose tissue, namely, perienteric, perinephric and epidydimal depots; (3) deregulation of glucose metabolism, characterized by increased glucose intolerance and fasting glycemia, and also alterations in insulin secretion, sensitivity, and signaling; (4) dyslipidemia characterized by changes on lipid profile; (5) increased deposition of lipids in the liver; (6) hypertension; and finally, (7) alterations in sympathetic nervous system activation (Table 3).

Comparison of the aforementioned disease models identified HF 19 weeks and the Zucker fatty rat (at 17 and 23 weeks) as presenting more alterations related to weight, fat and lipid metabolism, and blood pressure. Zucker 17 and 23 weeks also presented higher alterations in glucose metabolism and insulin secretion. Due to its highly disrupted lipid and glucose metabolism and a fasting glycemia levels above 126 mg/dl, the Zucker diabetic rat exhibits a phenotype that has been chosen by several research groups and pharmaceutical companies as an obesity and T2D model [27]. It is also notable that this genetic animal model of obesity and T2D is used as a sleep apnea model [27]. It is our view that the widespread use of this model should be questioned, given that: (1) The Zucker diabetic rat lacks functional leptin receptors (fa/fa genotype), because leptin receptor deficiency is infrequently seen in obese and T2D humans since leptin receptor mutations are relatively rare [28]; (2) we have shown here that Zucker diabetic rats do not exhibit an overactivation of the sympathetic nervous system, probably due to the absence of leptin receptors, which is generally accepted to be a principle driver of obesity and metabolic-related illness [14,15,29]. In short, this animal model does not reflect the phenotype and the pathophysiological mechanisms behind the metabolic dysfunction widely presented in humans.

In the last few years, an alternative path to the induction of obesity and/or T2D in animal models has been through dietary manipulation. However, in this respect, there is no consensus about the most appropriate diet (HF, HSu, HFHSu, HF plus low sucrose, low fat plus HSu, among others) and time of exposure to it. Herein, we showed that when compared to all other models studied, rats exposed to an HF diet for 19 weeks presented a phenotype most consistent with the human condition, inclusive of alterations in weight, fat amount, lipid metabolism, in insulin sensitivity and glucose tolerance, elevated sympathetic activity and blood pressure, and lipid deposition in the liver (Figure 1, Figure 2, Figure 3, Figure 4, Figure 5 and Figure 6). The only alterations that were more pronounced in another model (HFHSu 25 weeks) were insulin levels and C-peptide levels (Figure 2D), which are associated with endogenous pancreatic insulin secretion, suggesting that the “sugar” component of this diet likely has a higher impact on the pancreatic secretion of insulin. This is in agreement with the data observed in minipigs, where the long-term high fat-high sugar diet increased the expression levels of caspase-3, Bax, and insulin and decreased the expression levels of proliferating cell nuclear antigen and Bcl-2, indicating increased β-cell apoptosis [30].

That the HF19 weeks model is the model that exhibits more phenotypic characteristics similar to humans is in agreement with other studies, which have shown that the long-term administration of diets containing 40% to 60% of lipids promotes metabolic disorders and induces insulin resistance, arterial hypertension, and obesity in animal models and humans [31], T2D and hypertriglyceridemia [32], among others. In addition, we showed that with this HF 19 weeks rat model, it is possible to obtain the typical comorbidities of type 2 diabetes (Figure 4 and Figure 5), such as hypertension and increased lipid deposition typical of NAFLD [33]. In fact, although almost all hypercaloric regimens were able to induce a similar degree of hypertension, the HF19 weeks model was one of the animal models that presented with a high percentage of lipid deposition in the liver similar to the Zucker fatty animal. However, since the Zucker diabetic fatty rats lack the overactivity of the sympathetic nervous system, we propose that the HF 19 weeks represents more closely the human condition.

Another common T2D model frequently used is the chemical induction with low doses of STZ together with dietary manipulation, which aims to achieve a high fasting glycemia due to the mild impairment of insulin secretion [34,35]. We showed that the HF+STZ group exhibits several metabolic characteristics of human obesity and T2D, such as increased weight gain, glucose intolerance, increased fasting glycemia, and hypertension (Figure 1, Figure 2, and Figure 4). However, we demonstrated for the first time that this model failed to develop insulin resistance, in agreement with the observed lack of hyperinsulinemia (Figure 2D), which is now known to be a major cause of this pathological feature. For many years, insulin resistance was considered the cause of hyperinsulinemia. However, it is now generally accepted that hyperinsulinemia can itself trigger insulin resistance, leading to a vicious cycle of hyperinsulinemia–insulin resistance (for a review, see [36]). In fact, several animal and human studies have supported the view that increased insulin levels, even in the presence of normal weight, induce insulin resistance, by producing insulin receptor desensitization [37,38,39]. Further, it was recently found in rat hepatoma cells that persistent high insulin levels are associated with a persistent alteration at the insulin receptor tyrosine kinase domain, which may act to initiate or progressively worsen insulin resistance [40]. Therefore, our data are in agreement with the fact that hyperinsulinemia can cause insulin resistance, given that the HF+STZ model studied here lacks increased insulin secretion (Figure 2D) and does not exhibit insulin resistance, despite the fact that other dysmetabolic features are present.

## 5. Conclusions

The incidence and prevalence of metabolic diseases such as obesity and T2D has increased dramatically. In order to understand the mechanisms involved, it is essential that we identify the most appropriate animal models, i.e., those which most closely mimic the metabolic and cardiovascular changes observed in humans. In the present investigation, we evaluated several hypercaloric rat models as well as a genetic rat model, the Zucker diabetic rat (leptin receptor knockout), with the aim achieving this goal. We concluded that the most appropriate rat model is the 19-week HF diet, which provides a better fit than the Zucker rat, with the Zucker rat representing the bets of the rest. In contrast to the Zucker rat, the 19-week HF diet has the advantage of better representing the wider phenotype presented in humans. Notable in this respect is the fact that the Zucker rat did not exhibit sympathetic hyperactivity. Additionally, we conclude that of all the animal models tested, the HF+STZ and HSu 4 weeks animals should be excluded from future studies relating to metabolic syndrome-related disorders, due to the absence in these models of insulin resistance and obesity/dyslipidemia.

## Figures and Tables

**Figure 1 nutrients-11-01197-f001:**
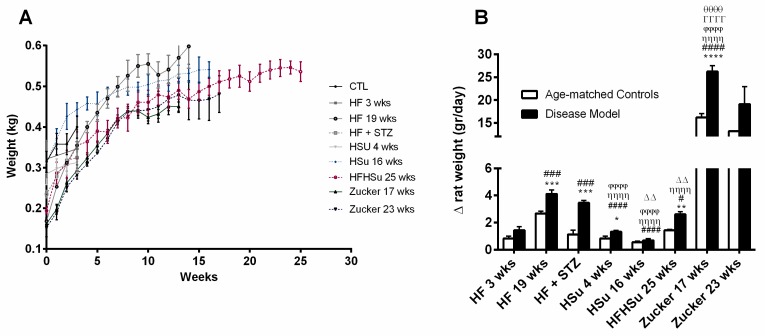
Weight gain in animal models of type 2 diabetes and obesity. (**A**) Growth curves. (**B**) Weight increase in animal groups expressed as g/day. All groups of hypercaloric diet animals and their aged-matched controls as well as Zucker diabetic rat and its lean controls of 17 weeks included 6–8 animals per group; Zucker diabetic rat of 23 weeks included 3 animals per group, and its lean control only one animal. Bars represent means ± SEM. Two-way ANOVA with Bonferroni’s multiple comparisons test: * *p* < 0.05, ** *p* < 0.01, *** *p* < 0.001, and **** *p* < 0.0001 comparing age-matched controls with disease models; ^#^
*p* < 0.05, ^###^
*p* < 0.001, and ^####^
*p* < 0.0001 comparing HF3 weeks with HF19 weeks, HF+STZ, HSu 4 weeks, HSu 4 weeks, HFHSu 25 weeks, and Zucker 17 weeks, respectively; ^ȠȠȠȠ^
*p* < 0.0001 comparing HF19 weeks with HSu 4 weeks, HSu 16 weeks, HFHSu 25 weeks, and Zucker 17 weeks; ^ϕϕϕϕ^
*p* < 0.0001 comparing HF+STZ with HSu 4 weeks, HSu 16 weeks, and Zucker 17 weeks; ^ΔΔ^
*p* < 0.01 comparing HSu 4 weeks with HSu 16 weeks and Zucker 17 weeks; ^θθθθ^
*p* < 0.0001 comparing HFHSu 25 weeks with Zucker 17 weeks; ^ΓΓΓΓ^
*p* < 0.0001 comparing HSu 16 weeks with Zucker 17 weeks.

**Figure 2 nutrients-11-01197-f002:**
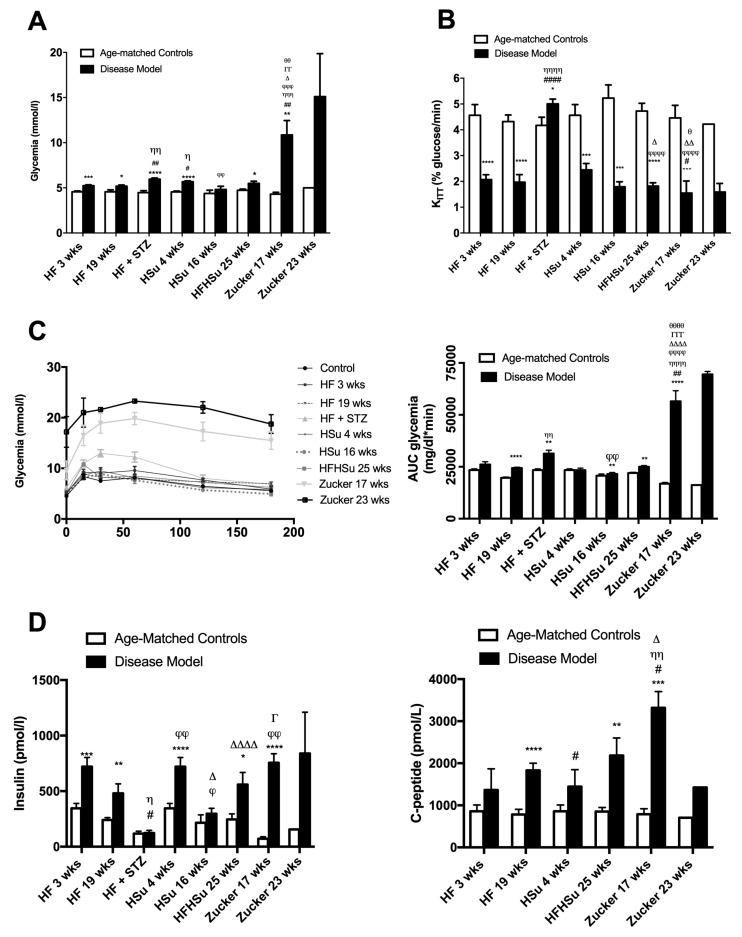
Glucose metabolism and insulin action and secretion in animal models of type 2 diabetes and obesity. (**A**) Fasting glycemia levels; (**B**) insulin sensitivity expressed as the constant of insulin tolerance test (K_ITT_); (**C**) glucose tolerance depicted as glucose excursion curves (**left panel**) and as the area under the curve (AUC) obtained from the glucose excursion curves (**right panel**); (**D**) fasting insulin (left panel) and C-peptide (right panel) levels in animal models of obesity and type 2 diabetes. All groups of hypercaloric diet animals and its aged-matched controls as well as Zucker diabetic rat and its lean controls of 17 weeks included 6–8 animals per group; Zucker diabetic rat of 23 weeks included 3 animals per group, and its lean control only one animal. Bars represent means ± SEM; two-way ANOVA with Bonferroni’s multiple comparisons test: * *p* < 0.05, ** *p* < 0.01, *** *p* < 0.001, and **** *p* < 0.0001 comparing age-matched controls with disease models; ^#^
*p*< 0.05, ^##^
*p* < 0.01, and ^####^
*p* < 0.0001 comparing HF 3 weeks group with HF 19 weeks, HF+STZ, HSu 4 weeks, and Zucker 17 weeks; ^Ƞ^
*p* < 0.05, ^ȠȠ^
*p* < 0.01, ^ȠȠȠ^
*p* < 0.001, and ^ȠȠȠȠ^
*p* < 0.0001 comparing HF 19 weeks with HF+STZ, HSu 4 weeks, HFHSu 25 weeks, and Zucker 17 weeks; ^ϕ^
*p* < 0.05, ^ϕϕ^
*p* < 0.01, ^ϕϕϕ^
*p* < 0.001, and ^ϕϕϕϕ^
*p* < 0.0001 comparing HF+STZ with HSu 16 weeks, HFHSu 25 weeks, and Zucker 17 weeks; ^∆^
*p* < 0.05; ^∆∆^
*p* < 0.01, and ^∆∆∆∆^
*p* < 0.0001 comparing HSu 4 weeks with HFHSu 25 weeks and Zucker 17 weeks; ^Γ^
*p* < 0.05, ^ΓΓ^
*p* < 0.01, and ^ΓΓΓ^
*p* < 0.001 comparing HSu 16 weeks group with Zucker 17 weeks; ^θ^
*p* < 0.05, ^θθ^
*p* < 0.01, and ^θθθθ^
*p* < 0.0001 comparing HFHSu 25 weeks with Zucker 17 weeks.

**Figure 3 nutrients-11-01197-f003:**
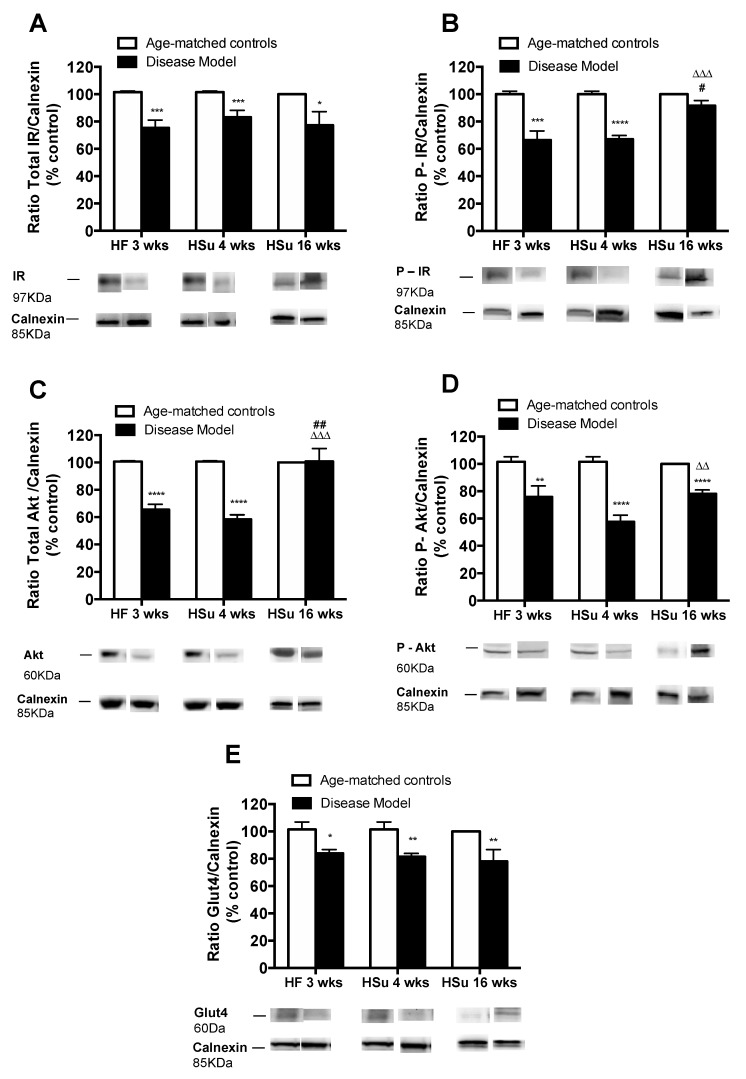
Alterations in the expression of proteins involved in insulin signaling in insulin-sensitive tissues in animal models of obesity and type 2 diabetes. (**A**) Total insulin receptor (IR) expression (**B**) phosphorylated IR expression; (**C**) total AKT expression; (**D**) phosphorylated AKT expression; (**E**) Glu4 expression. Below the graphs, representative bands from the Western blots for the correspondent proteins are shown. Bars represent means ± SEM; one- and two-way ANOVA with Bonferroni’s multiple comparisons test: * *p* < 0.05, ** *p* < 0.01, *** *p* < 0.001, and **** *p* < 0.0001 comparing age-matched controls with disease models; ^#^
*p* < 0.05 and ^##^
*p* < 0.01 comparing HF 3 weeks with HSu 16 weeks; ^ΔΔ^
*p* < 0.01 and ^ΔΔΔ^
*p* < 0.001 comparing HSu 4 weeks with HSu 16 weeks.

**Figure 4 nutrients-11-01197-f004:**
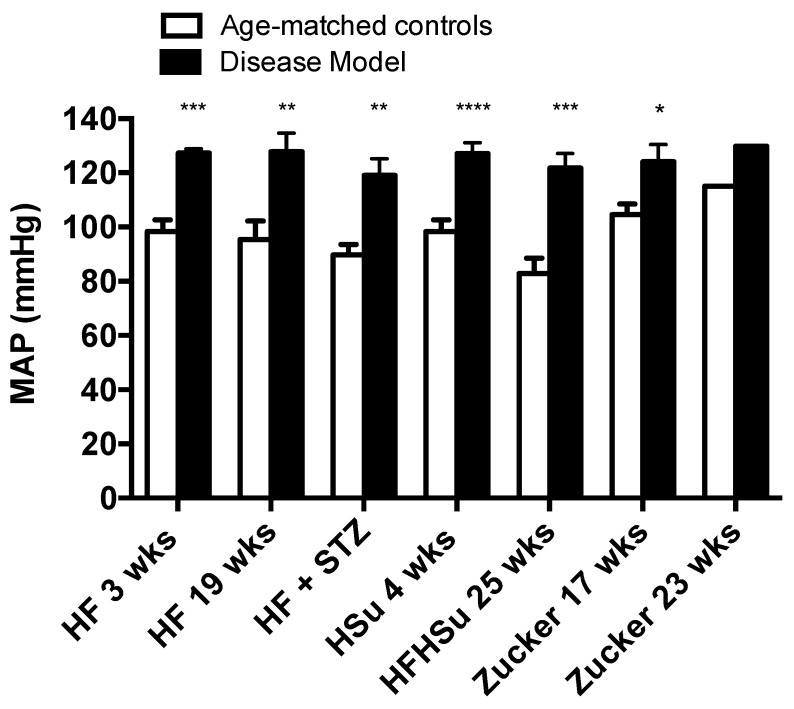
Mean arterial pressure levels in the distinct models of obesity and type 2 diabetes. All groups of hypercaloric diet animals and its aged-matched controls as well as Zucker diabetic rat and its lean controls of 17 weeks included 6–8 animals per group; Zucker diabetic rat of 23 weeks included 3 animals per group, and its lean control only one animal. Bars represent means ± SEM; two-way ANOVA with Bonferroni’s multiple comparisons test: * *p* < 0.05, ** *p* < 0.01, *** *p* < 0.001, and **** *p* < 0.0001 comparing age-matched controls with disease models.

**Figure 5 nutrients-11-01197-f005:**
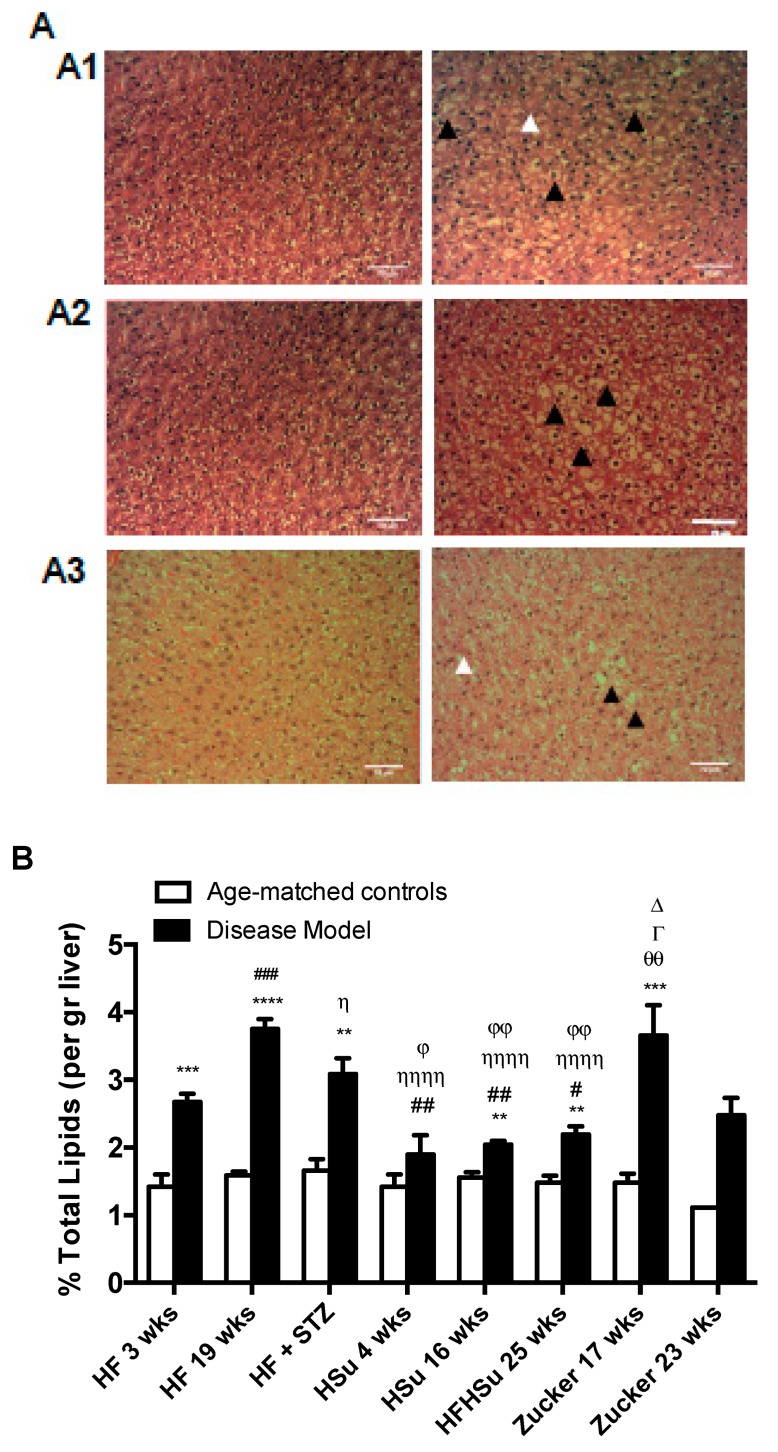
Lipid deposition in the liver in the distinct models of obesity and type 2 diabetes. (**A**) Hematoxicilin and eosin images of liver slices from HF 19 weeks (A1), HFHSu 25 weeks (A2), and Zucker 23 weeks (A3) and their respective controls. (**B**) Percentage of lipid deposition in the liver quantified by the Folch method [26]. All groups of hypercaloric diet animals and its age-matched controls as well as Zucker diabetic rat and its lean controls of 17 weeks included 6–8 animals per group; Zucker diabetic rat of 23 weeks included 3 animals per group, and its lean control only one animal. Bars represent means ± SEM; two-way ANOVA with Bonferroni’s multiple comparisons test: ** *p* < 0.01, *** *p* < 0.001, and **** *p* < 0.0001 comparing age-matched controls with disease models; ^#^
*p* < 0.05, ^##^
*p* < 0.01, and ^###^
*p* < 0.001 comparing HF 3 weeks with HF 19 weeks, HSu 4 weeks, and HFHSu 25 weeks; ^Ƞ^
*p* < 0.05 and ^ȠȠȠȠ^
*p* < 0.0001 comparing HF 19 weeks with HF+STZ, HSu 4 weeks, and HFHSu 25 weeks; ^ϕ^
*p* < 0.05, ^ϕϕ^
*p* < 0.01 comparing HF+STZ with HSu 4 weeks and HFHSu 25 weeks; ^∆^
*p* < 0.05 comparing HSu 4 weeks with Zucker 17 weeks; ^Γ^
*p* < 0.05 comparing HSu 16 weeks with Zucker 17 weeks; ^θθ^
*p* < 0.01 comparing HFHSu 25 weeks with Zucker 17 weeks. Black arrowheads represent lipidosis and white arrowheads fibrosis. Scale bar, 100 μm; original magnification, 20×.

**Figure 6 nutrients-11-01197-f006:**
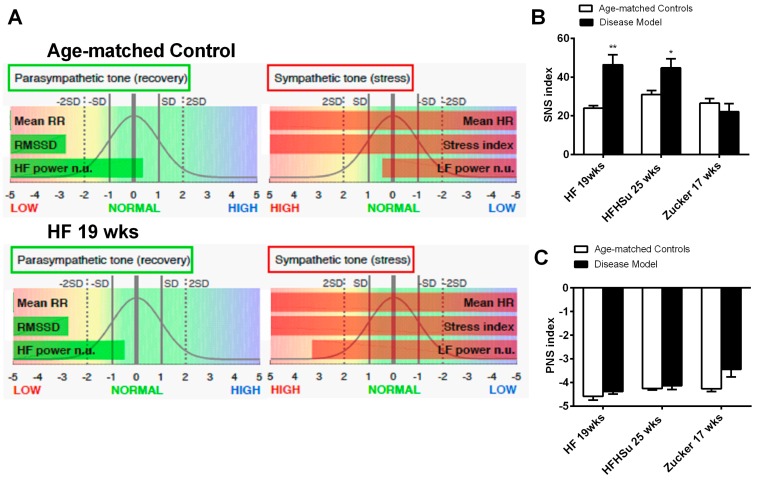
Effect of hypercaloric diets and genetic deletion of leptin receptors on sympathetic activity evaluated by spectral analysis of the heart rate. (**A**) shows representative experiments of power spectral density (PSD) calculated in control, in HF 3 weeks, and in HF+ STZ animals. (**B**,**C**) Autonomic function assessed by the ratio between the percentage of low frequencies (LF) that represents the sympathetic component of the autonomic nervous system and the percentage of high frequencies (HF) that represents the parasympathetic component of the autonomic nervous system. Frequencies are presented in normalized units. All groups of hypercaloric diet animals and their aged-matched controls as well as Zucker diabetic rat and its lean controls of 17 weeks included 6–8 animals per group; Zucker diabetic rat of 23 weeks included 3 animals per group, and its lean control only one animal. Bars represent means ± SEM; two-way ANOVA with Bonferroni’s multiple comparisons test: * *p* < 0.05 and ** *p* < 0.01 comparing age-matched controls with disease models.

**Table 1 nutrients-11-01197-t001:** Adipose tissue depots and lipid profile (total cholesterol, triacylglycerols, high-density lipoprotein (HDL)–cholesterol and low-density lipoprotein (LDL)–cholesterol) in animal models of obesity and type 2 diabetes.

		Total Fat (g/kg Body Weight)	Perienteric Fat (g/kg Body Weight)	Epidydymal Fat (g/kg Body Weight)	Perinephric Fat (g/kg Body Weight)	Cholesterol (mmol/L)	HDL-Cholesterol (mmol/L)	LDL-Cholesterol (mmol/L)	Triglycerides (mmol/L)
**HF 3 weeks**	age-matched control	48.72 ± 2.21	9.56 ± 0.43	19.81 ± 2.06	17.80 ± 1.72	1.77 ± 0.09	0.72 ± 0.03	0.13 ± 0.01	0.35 ± 0.04
	disease model	70.05 ± 3.62 ***	13.82 ± 0.79 ***	38.47 ± 0.97 **	30.18 ± 1.97 **	1.83 ± 0.08	0.53 ± 0.03 ***	0.11 ± 0.02	0.51 ± 0.04 **
**HF 19 weeks**	age-matched control	91.23 ± 7.63	13.76 ± 1.60	29.66 ± 1.98	29.01 ± 2.04	1.69 ± 0.08	0.80 ± 0.05	0.12 ± 0.01	1.14 ± 0.09
	disease model	223.37 ± 2.58 ^****,####^	32.12 ± 2.23 ^****,####^	61.58 ± 2.08 ^****,##,ΔΔΔΔ^	75.21 ± 2.17 ^****,####,ΔΔΔΔΔ^	2.48 ± 0.15 ^***,###^	0.55 ± 0.03 ^**,###^	0.23 ± 0.01 ^****,###^	1.66 ± 0.27 ^*,###^
**HF+** **STZ**	age-matched control	54.23 ± 2.81	11.00 ± 0.88	18.66 ± 1.15	25.27 ± 1.42	-	-	-	-
	disease model	61.37 ± 2.50 ^ηηηη^	13.21 ± 0.62 ^ηηηη^	21.83 ± 0.88 ^*,####,ηηηη^	26.33 ± 1.89 ^ηηηη^	-	-	-	-
**HSu 4 weeks**	age-matched control	48.72 ± 2.21	9.56 ± 0.43	19.81 ± 2.06	17.80 ± 1.72	1.77 ± 0.09	0.72 ± 0.03	0.13 ± 0.01	0.35 ± 0.04
	disease model	52.46 ± 2.28 ^###,ηηηη,ϕϕ^	10.17 ± 0.52 ^##,ηηηη,ϕϕ^	20.62 ± 2.34	23.37 ± 0.95 ^*^	2.13 ± 0.11 ^*,#^	0.69 ± 0.05 ^##^	0.14 ± 0.02 ^ηη^	0.61 ± 0.07 ^**,###,ηη^
**HSu 16 weeks**	age-matched control	59.83 ± 1.74	13.58 ± 0.31	25.98 ± 1.35	20.27 ± 2.11	1.80 ± 0.10	0.79 ± 0.04	0.10 ± 0.01	0.48 ± 0.06
	disease model	80.44 ± 2.34 ^****,ηηηη,ΔΔΔΔ^	17.93 ± 0.75 ^***,ηηη,ΔΔΔΔ^	31.43 ± 2.40 ^ηηηη,ϕϕϕ,Δ^	31.08 ± 2.66 ^*,ηηηη^	1.98 ± 0.11 ^η^	0.60 ± 0.04 ^**,####^	0.12 ± 0.01 ^ηηη^	0.79 ± 0.05 ^**,ηη^
**HFHSu 25 weeks**	age-matched control	66.44 ± 3.93	15.58 ± 0.82	25.08 ± 1.52	25.78 ± 1.75	1.90 ± 0.25	0.63 ± 0.04	0.24 ± 0.03	0.90 ± 0.14
	disease model	113.91 ± 9.72 ^***,###,ηηηη,Γ^	24.44 ± 2.84 ^**,###,ϕϕϕ,ΔΔΔ^	36.90 ± 2.83 ^**,ηηηη,ϕϕϕ,ΔΔ^	52.57 ± 4.45 ^****,#,ηη,ΓΓ^	2.26 ± 0.16	0.59 ± 0.04 ^η^	0.37 ± 0.05 ^###,ΔΔ,ΓΓΓΓ^	1.53 ± 0.13 ^**,####^
**Zucker 17 weeks**	age-matched control	42.83 ± 4.15	6.55 ± 0.65	8.82 ± 0.71	11.37 ± 0.92	2.27 ± 0.07	1.64 ± 0.13	0.10 ± 0.02	1.16 ± 0.21
	disease model	151.77 ± 7.58 ^****,####,ηηηη,ΓΓΓΓ,θ^	15.11 ± 1.25 ^***,ηηη,ΔΔ^	22.75 ± 3.15 ^**,#,ηηηη,θθ^	35.05 ± 1.80 ^****,ηηηη,ΔΔΔΔ^	4.51 ± 0.29 ^***,θθθθ,^ ^ηηη,ΔΔΔΔ^	0.87 ± 0.03 ^***,####^	0.70 ± 0.07 ^****,####^	8.72 ± 1.21 ^***,####,ΓΓΓΓ,θθθθ^
**Zucker 23 weeks**	age-matched control	30.44 ± 0.00	3.19 ± 0.00	7.45 ± 0.00	6.33 ± 0.00	-	-	-	-
	disease model	168.3 ± 34.94	12.27 ± 2.47	26.33 ± 4.21	45.64 ± 6.59	-	-	-	-

All groups of hypercaloric diet animals and its aged-matched controls as well as Zucker diabetic rat and its lean controls of 17 weeks included 6–8 animals per group; Zucker diabetic rat of 23 weeks included 3 animals per group, and its lean control only one animal. Values represent means ± SEM; one- and two-way ANOVA with Bonferroni’s multiple comparisons test: * *p* < 0.05, ** *p* < 0.01, *** *p* < 0.001, and **** *p* < 0.0001 comparing age-matched controls with disease models; ^#^
*p* < 0.05, ^##^
*p* < 0.01, ^###^
*p* < 0.001, and ^####^
*p* < 0.0001 comparing HF 3 weeks group with HF 19 weeks, HF+STZ, HSu 4 weeks, HSu 16 weeks, HFHSu 25 weeks, and Zucker 17 weeks; ^Ƞ^
*p* < 0.05, ^ȠȠ^
*p* < 0.01, ^ȠȠȠ^
*p* < 0.001, and ^ȠȠȠȠ^
*p* < 0.0001 comparing HF 19 weeks with HF+STZ, HSu 4 weeks, HSu 16 weeks, HFHSu 25 weeks, and Zucker 17 weeks; ^ϕ^
*p* < 0.05, ^ϕϕ^
*p* < 0.01, ^ϕϕϕ^
*p* < 0.001, and ^ϕϕϕϕ^
*p* < 0.0001 comparing HF+STZ with HSu 4 weeks, HSu 16 weeks, HFHSu 25 weeks, and Zucker 17 weeks; ^∆^
*p* < 0.05, ^∆∆^
*p* < 0.01, ^∆∆∆^
*p* < 0.001, and ^∆∆∆∆^
*p* < 0.0001 comparing HSu 4 weeks with HSu 16 weeks, HFHSu 25 weeks, and Zucker 17 weeks; ^θ^
*p* < 0.05, ^θθ^
*p* < 0.01, and ^θθθθ^
*p* < 0.0001 comparing HFHSu 25 weeks with Zucker 17 weeks; ^Γ^
*p* < 0.05, ^ΓΓ^
*p* < 0.01, and ^ΓΓΓΓ^
*p* < 0.0001 comparing HSu 16 weeks with HFHSu 25 weeks and Zucker 17 weeks.

**Table 2 nutrients-11-01197-t002:** Catecholamine (epinephrine and norepinephrine) levels in plasma and in adrenal medulla in animal models of obesity and type 2 diabetes.

		Plasma NE + Epi (pmol/mL)	Adrenal Medulla NE+Epi (pmol/mg Tissue)
**HF 3 weeks**	age-matched control	52.01 ± 6.54	36038.33 ± 2300.22
	disease model	107.45 ± 8.17 ****	50954.94 ± 4604.37 **
**HF 19 weeks**	age-matched control	80.50 ± 13.75	39048.39 ± 3164.09
	disease model	161.10 ± 36.80 *	51395.94 ± 2612.16 **
**HSu 4 weeks**	age-matched control	52.01 ± 6.54	36038.33 ± 2300.22
	disease model	142.48 ± 18.21 ****	57081.27 ± 5877.55 **
**HSu 16 weeks**	age-matched control	130.40 ± 29.74	-
	disease model	159.88 ± 51.57	-

Values represent means ± SEM; one- and two-way ANOVA with Bonferroni’s multiple comparisons test: * *p* < 0.05, ** *p* < 0.01, and **** *p* < 0.0001 comparing age-matched controls with disease models.

**Table 3 nutrients-11-01197-t003:** Summary of the main pathological features of animal models of dysmetabolism.

	Obesity	Increased Fasting Glycemia	Insulin Resistance	Glucose Intolerance	Hyperinsulinemia	Hyper Cholesterolemia	Hyper Triglyceridemia	Lipid Deposition in the Liver	Alterations in Insulin Signaling, in White Adipose Tissue	Increased Catecholamine Levels/SNS Activity
**HF 3 weeks**	**x**	**✓**	**✓**	**x**	**✓**	**x**	**✓**	**✓**	**✓**	**✓**
**HF 19 weeks**	**✓**	**✓**	**✓**	**✓**	**✓**	**✓**	**✓**	**✓**	****-****	**✓**
**HF + STZ**	**✓**	**✓**	**x**	**✓**	**x**	****-****	****-****	**✓**	****-****	**-**
**HSu 4 weeks**	**✓**	**✓**	**✓**	**x**	**x**	**✓**	**✓**	**x**	**✓**	**✓**
**HSu 16 weeks**	**x**	**x**	**✓**	**x**	**x**	**x**	**✓**	****-****	**✓**	**-**
**HFHSu 25 weeks**	**x**	**✓**	**✓**	**✓**	**✓**	**x**	**✓**	**✓**	****-****	**✓**
**Zucker 17 weeks**	**✓**	**✓**	**✓**	**✓**	**✓**	**✓**	**✓**	**✓**	****-****	**-**
**Zucker 23 weeks**	**✓**	**✓**	**✓**	**✓**	**✓**	****-****	****-****	**✓**	****-****	**-**

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
