# Peer review of "Evaluating the Impact of Different Hypercaloric Diets on Weight Gain, Insulin Resistance, Glucose Intolerance, and its Comorbidities in Rats"

_nutrients, 2019, doi:10.3390/nu11061197_

Reviewer 1 Report

Reviewer’s comments and suggestions for Authors

In this research manuscript, the authors have evaluated the impact of different hypercaloric diets on several parameters such as weight gain, insulin resistance, glucose intolerance and associated comorbidities in animal models. Basically, they aimed a realistic selection of the appropriate models of metabolic diseases. For that the authors choose different diets specifically the hypercaloric  diets (high fat (HF), high sucrose (HSu), high fat plus high sucrose (HFHSu) and high fat plus streptozotocin (HF+STZ)) during diverse exposure times (HF 3wks, HF 19 weeks, HSu 4 weeks, HSu  16 weeks, HFHSu 25 weeks, HF3 weeks + STZ) and a genetic model, the Zucker fatty rat. The study ended-up in result for the animal models that exhibited a higher weight gain in Zucker fatty rat and the rats treated with HF during 19 weeks, representing this last model a more realistic phenotypic variety present in humans. Moreover, all animal models exhibited insulin resistance and dyslipidemia, except for the HF+STZ and HSu 4weeks.

 The manuscript is well written in terms of the English language. However, there few comments and advise to modify in the current version of the manuscript and submit again as a revised version.

 (1) In the abstract part, the authors have suggested that they conclude that none of the animal models tested gathers all phenotypic features and comorbidities of type 2 diabetes present and none can be used a reliable model. I think there were lots of paper previously published animal model which were suggested that HFD model can be used for the experiments related to type 2 diabetes. In my opinion, the author needed to rewrite the sentence and modify it based on his current results.

   (2) There may be some formula to calculate to mean blood pressure, you need to add in the section blood pressure evaluation. In Figure 3, there is Mean arterial pressure, is it same as mean blood pressure?

 (3) As seen in the section for Western blot analysis, I think the procedure is not fully described. I would suggest the author include all steps for doing this analysis.

 (4) The authors have to clearly mention the idea of evaluating the autonomic nervous system in the paper, as introduction does not have any pre-details about it.

 (5) No need to repeat the line both in the introduction and discussion part “with the induction of obesity and/or T2D being achieved by surgical, chemical or dietary manipulations, or by the combination of these techniques” the author has already mentioned in the introduction part.

(6) Please explain for the result for the figure D “we describe for the first time that this model failed to develop insulin resistance, being this in agreement with the absence of hyperinsulinemia (Figure 2D), a major cause for the development of this pathological feature”.

 (7) The western blot results should be added in the main manuscript as a figure, not in the supplementary file (relevant). These results should be described well in the discussion part as well. If the author has the photos of the complete gel, then those figures have to been removed and pasted a complete gel for each signaling markers. That would be better.

 (8) No need to add the DOI as the authors have not added it for all. Reference 6, 9... and so on. (9) Please check the format of reference number 11, 14, 16,

 Author Response

We would like to thank the reviewer the comments on our manuscript, which recognize the quality of our work. We have made improvements in the manuscript following the reviewer suggestions and we hope that this new version of the manuscript is an acceptable form to be published in Nutrients. The present version of the manuscript was reviewed by an English Native, Professor Mark Evans from the Centres for Discovery Brain Sciences and Cardiovascular Sciences, University of Edinburgh.

(1) In the abstract part, the authors have suggested that they conclude that none of the animal models tested gathers all phenotypic features and comorbidities of type 2 diabetes present and none can be used a reliable model. I think there were lots of paper previously published animal model which were suggested that HFD model can be used for the experiments related to type 2 diabetes. In my opinion, the author needed to rewrite the sentence and modify it based on his current results.

The reviewer is right and we the animal model in our study that gathers more phenotypic as well as causal characteristics of diabetes is the HF animal. We changed our manuscript accordingly and now it can be read in the conclusion:  “In this work, we evaluated several hypercaloric rat models as well as a genetic rat model, obtained by the deletion of leptin receptors, aiming to be able in the future to select more accurately the most appropriate models for study obesity, prediabetes, metabolic syndrome and T2D. We suggest that the most appropriate rat models to study obesity are the Zucker and the models submitted to HF diet. However, in contrast to genetically manipulated animals, the dietary models have the advantage of representing the phenotypic variety present in humans. In fact, we showed that all animals exhibit sympathetic hyperactivity, which is a causal factor of metabolic disease, except the Zucker diabetic rat. Additionally, we conclude that all the animal models tested in the present work can be used as metabolic syndrome animal models, except the HF+STZ and HSu 4 wks animals, due to the absence of insulin resistance and obesity/dyslipidemia, respectively. Furthermore, we conclude that the animal model tested that gathers the majority of the phenotypic features and comorbidities of T2D present in humans is the HF19 wks. Therefore, we suggest that for the study of metabolic diseases the animal model must be chosen taking into account the specific research requirements.”

The abstract was also corrected.

 (2) There may be some formula to calculate to mean blood pressure, you need to add in the section blood pressure evaluation. In Figure 3, there is Mean arterial pressure, is it same as mean blood pressure?

Yes, we used indistinctly mean arterial pressure and mean blood pressure. We replaced mean blood pressure by mean arterial pressure throughout the manuscript. Mean arterial pressure was calculated using the values of systolic blood pressure (SBP) and diastolic blood pressure (DBP) by the Iox 2.9.5.73 software, Emka Technologies. To calculate mean arterial pressure, double the diastolic blood pressure and add the sum to the systolic blood pressure. Then divide by 3.  MAP = (SBP + 2 (DBP))/3. We added this information to methods section.

 (3) As seen in the section for Western blot analysis, I think the procedure is not fully described. I would suggest the author include all steps for doing this analysis.

 A full description was added to this section.

(4) The authors have to clearly mention the idea of evaluating the autonomic nervous system in the paper, as introduction does not have any pre-details about it.

 We have included a paragraph in the introduction section in where we make the link between metabolic diseases and autonomic dysfunction, as this last pathological feature has been pointed as one of the causal mechanisms of metabolic diseases. Now it can be read: “One of the mechanisms involved in the pathogenesis of metabolic diseases and its comorbidities is autonomic dysfunction. Obesity and other metabolic diseases, as type 2 diabetes and metabolic syndrome are associated with an increased sympathetic and decreased parasympathetic peripheral activity [11,12] with previous studies showing an increase of serum noradrenaline, noradrenaline excretion, renal and heart NA-SR, muscle sympathetic nervous system activity and heart rate variability indexes in obese, metabolic syndrome and type 2 diabetes individuals [13-16]. Autonomic dysfunction has been linked to target organ damage, such as diastolic dysfunction, ventricular hypertrophy or cardiac remodelling as well as with insulin resistance, hyperinsulinemia and dyslipidemia [13,15]. It has been postulated that prevention of autonomic dysfunction could be a target to reduce these comorbidities and that the assessment of autonomic function could be used as an early biomarker for metabolic diseases [17-18].”

 (5) No need to repeat the line both in the introduction and discussion part “with the induction of obesity and/or T2D being achieved by surgical, chemical or dietary manipulations, or by the combination of these techniques” the author has already mentioned in the introduction part.

We changed the sentence.

(6) Please explain for the result for the figure D “we describe for the first time that this model failed to develop insulin resistance, being this in agreement with the absence of hyperinsulinemia (Figure 2D), a major cause for the development of this pathological feature”.

We included a paragraph in the discussion section where we explain the result from figure 2D. Now it can be read: “However, we describe for the first time that this model failed to develop insulin resistance, being this in agreement with the absence of hyperinsulinemia (Figure 2D), a major cause for the development of this pathological feature. During several years insulin resistance was considered the cause of hyperinsulinemia, however nowadays it accepted that hyperinsulinemia can itself tether insulin resistance leading to a vicious cycle of hyperinsulinemia-insulin resistance [for a review see 37]. In fact, several animal and human studies have supported this last notion and showed that increased insulin levels even in the presence of normal weight induce insulin resistance, by producing insulin receptor desensitization [38-40]. Also, it was recently found in rat hepatoma cells that persistent high insulin levels are associated with a persistent alteration at the insulin receptor tyrosine kinase domain that may act to initiate or progressively worsen insulin resistance [41]. Therefore, our data is in agreement the that hyperinsulinemia can cause insulin resistance since the HF+STZ model that lacks increased insulin secretion (figure 2D) do not exhibit insulin resistance but present other dysmetabolic features.”

 (7) The western blot results should be added in the main manuscript as a figure, not in the supplementary file (relevant). These results should be described well in the discussion part as well. If the author has the photos of the complete gel, then those figures have to been removed and pasted a complete gel for each signaling markers. That would be better.

We included the figure with adipose tissue western blots in the manuscript (now figure 3) and include the complete gels of the western blots as supplemental data.

 (8) No need to add the DOI as the authors have not added it for all. Reference 6, 9...and so on. (9) Please check the format of reference number 11, 14, 16,

We corrected all the references as suggested by the reviewer. We acknowledge the reviewer the attention given to our manuscript.

Reviewer 2 Report

The study by Melo et al describes the use of different animal models to identify one that would be suitable to study type 2 diabetes and closely mimic the human scenario. My major comments regarding this study can be found below:

Earlier, models using high fat high sucrose diets to induce diabetes like phenotype have been described before. Please check the following references as few examples: Amor et al, 2019; Rasool et al, 2018; Burchfield et al, 2018, Sato et al, 2010. The study does not therefore advance our knowledge significantly in this context. Also, the introduction lacks mentioning previous studies that have used similar models.

Only a table has been provided with weights of WAT (White Adipose Tissue) for the different experimental groups. Representative images should be provided for each experimental group.

Please provide representative histological images of liver sections stained for lipid/fat deposition e.g. Oil red stainings to support the conclusions derived from the bar graphs

Overall, thus the study fails to significantly advance our understanding in this field.

Author Response

The study by Melo et al describes the use of different animal models to identify one that would be suitable to study type 2 diabetes and closely mimic the human scenario. My major comments regarding this study can be found below:

Earlier, models using high fat high sucrose diets to induce diabetes like phenotype have been described before. Please check the following references as few examples: Amor et al, 2019; Rasool et al, 2018; Burchfield et al, 2018, Sato et al, 2010. The study does not therefore advance our knowledge significantly in this context. Also, the introduction lacks mentioning previous studies that have used similar models.

We are aware that different hypercaloric diets have been tested in the context of metabolic disease, hyperlipidemic, hyperglucidic and the combination of both. Also, we believe that although the subject of our study might not be extremely original this paper contributes to advance our knowledge in the field as we tested different hypercaloric diets and different times of exposure to the diets and compared with the genetic model used as reference in the field “the Zucker diabetic rat”. Additionally, we evaluated not only the metabolic features of the disease, but also other components that are now known to be causes (autonomic dysfunction) and consequences (hypertension, lipid deposition in the liver as an indication of NAFLD) of the disease. We believe that this manuscript by testing different conditions might will be useful in future as a reference manuscript in the field of animal models for metabolic disease research, enabling time saving as well as the reduction in unnecessary costs.

Following the suggestions of the reviewer we included some references regarding the use of combined hyperlipidemic/hyperglucidic diet. However, we should also refer that the majority of the studies using combined hyperlipidemic/hyperglucidic diets were made in mice and not in rats as the one that we are submitting. Also, some of the studies using these diets do not evaluate the same parameters we evaluate in the present manuscript (for a review see Rasool et al. 2018).

Now in the introduction it can be read “Thus, hypercaloric, hyperlipidemic diets or the combination of both have been used to induce obesity and T2D, as well as metabolic syndrome, in animals [19-22]. However, a lack of consensus exists on the best diets and time of exposure to the diets used to promote alterations in metabolic parameters, as insulin resistance, hyperinsulinemia and glucose intolerance, among others, as well as to induce metabolic comorbidities as hypertension and non-alcoholic fatty liver disease.

Only a table has been provided with weights of WAT (White Adipose Tissue) for the different experimental groups. Representative images should be provided for each experimental group.

We are aware that it would be interesting to see the changes in the size of adipocytes with the different diet treatments, however we do not have samples from all the groups of animals and therefore we are not able to perform this. However, it is already described that HF and combined HFHSu diets as well as  Zucker fatty diabetic rats exhibit  increased adipocyte size (Poret et al. Int J Obes 2018; Gao et al. Plos One; 2015; Vasselli et al., Am J Physiol. 1992, among others).

Please provide representative histological images of liver sections stained for lipid/fat deposition e.g. Oil red stainings to support the conclusions derived from the bar graphs

We have included representative hematoxicilin  & eosin images of the livers from the animal models that present major lipid deposition (HF19wks, HFHSu and Zucker) and respective controls to support the conclusions derived from the bar graphs. However, we want to highlight that the lipid deposition in the liver how is presented was evaluated by a biochemical method described by Folch (Folch J, Lees M, Stanley GHS. A simple method for the isolation and purification of total lipides from animal tissues. Journal of Biological Chemistry 1957; 226 :497–509), where the lipid are extracted from the liver and at the end weighted.  The images support the conclusions obtained using the Folch method, but also show liver fibrosis in some of the animal models.

Overall, thus the study fails to significantly advance our understanding in this field.

As we previously stated we believe that this paper contributes to advance our knowledge in the field as we have tested different hypercaloric diets and different times of exposure to the diets and compared with the genetic model used as reference in the field “the Zucker diabetic rat”. Additionally, we have evaluated not only the metabolic features of the disease, but also other components that are now known to be causes (autonomic dysfunction) and consequences (hypertension, lipid deposition in the liver as an indication of NAFLD) of the disease. We believe that this manuscript by testing different conditions might will be useful in future as a reference manuscript in the field of animal models for metabolic disease research, enabling time saving as well as the reduction in unnecessary costs.

 Round  2

Reviewer 2 Report

Although the study does not rank high in the originality of its content, it provides an assessment of animal models used in metabolic research.

The major concerns raised previously have been addressed by the authors.